# MotionRL: Aligning Text-to-Motion Generation to Human Preferences with Multi-Reward Reinforcement Learning

## Abstract

We introduce **MotionRL**, the first approach to utilize Multi-Reward Reinforcement Learning for optimizing text-to-motion generation tasks and aligning them with human preferences. Previous works focused on improving numerical performance metrics on the given datasets, often neglecting the variability and subjectivity of human feedback. In contrast, our novel approach uses reinforcement learning to fine-tune the motion generator based on human preferences prior knowledge of the human perception model, allowing it to generate motions that better align human preferences. In addition, MotionRL introduces a novel multi-objective optimization strategy to approximate Pareto optimality between text adherence, motion quality, and human preferences. Extensive experiments and user studies demonstrate that MotionRL not only allows control over the generated results across different objectives but also significantly enhances performance across these metrics compared to other algorithms.

## 1 Introduction

High-quality human motion is in high demand across various fields, such as animation, robotics, and gaming. Due to the simplicity and user-friendliness of text input, text-driven human motion generation (Zhu et al., 2023; Aliakbarian et al., 2020; Bouazizi et al., 2022; Barsoum et al., 2018; Mao et al., 2019; Habibie et al., 2017; Yan et al., 2018; Aristidou et al., 2022; Lee et al., 2019)has become a prominent research topic in recent years. However, generating motion from text is a highly challenging multimodal task, and current methods face several issues. First, there is often a semantic mismatch between text and motion, where the algorithm must understand diverse textual descriptions and accurately generate corresponding motions. Second, the quality of generated motion needs to be as close to the ground truth as possible. Additionally, as a branch of generative tasks, human perception of motion is also highly important (Voas et al., 2023).

To address this task, traditional generative frameworks, such as Variational Autoencoders (VAEs) (Van Den Oord et al., 2017), have been used to align textual descriptions with motion sequences. Recently, more advanced methods like generative masked modeling (Guo et al., 2023) have aimed to model conditional masked motion to achieve better alignment between motion and text. Another approach involves Diffusion Models Dabral et al. (2023); Zhang et al. (2023b); Tevet et al. (2022); Chen et al. (2023), which predict motion sequences through a gradual denoising process. While these methods have shown promising results on traditional metrics, they suffer non-trivial limitations. On one hand, they require the motion length as input to guide generation, which may lead to motion quality degradation, since the length is closely tied to the content of the motion sequence (Pinyoanuntapong et al., 2024). On the other hand, predicting entire motion sequences works well for short texts, but results in inaccuracies for more complex textual inputs, particularly in the fine-grained details of the motion.

On the other hand, generative transformer-based methods (Jiang et al., 2023; Zhang et al., 2023a; Mao et al., 2024) alleviate some of the above issues by following the GPT-type training process, where the motion sequence is generated through autoregressive next-token prediction.

Regardless of the generation method, almost all mainstream research has largely ignored the role of human perception in evaluating generated motions. Generally, generating realistic human motion,

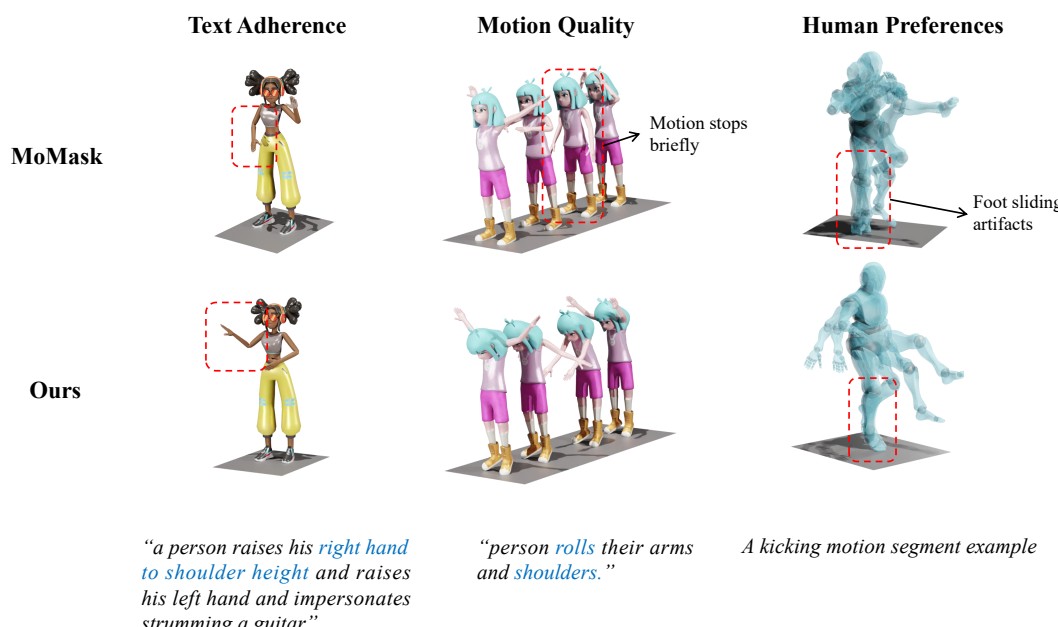

**Text Adherence**  **Motion Quality**  **Human Preferences**

*"a person raises his right hand to shoulder height and raises his left hand and impersonates strumming a guitar"*

*"person rolls their arms and shoulders."*

*A kicking motion segment example*

Figure 1: **Examples generated by MoMask and Ours.** Our method significantly outperforms the previous state-of-the-art MoMask in text adherence, motion quality and human preferences.

including smooth and natural movement, is more important than fitting existing error-based metrics, such as FID and R-Precision (Heusel et al., 2017). Current methods tend to overfit to small datasets and focus on optimizing for FID scores. In fact, experiments in Pinyoanuntapong et al. (2024) have shown that a better FID score does not necessarily correlate with human perception, as issues like visual artifacts and foot sliding are not well captured by these metrics. Since such artifacts are difficult to measure using existing metrics (Zhang et al., 2023b), human perception of generated motions becomes crucial.

Despite this, few studies have taken human perception into account. While some motion perception models (Voas et al., 2023; Wang et al., 2024) do incorporate human perceptual priors, these priors are often more subtle and complex than traditional metrics. Directly incorporating these priors (Wang et al., 2024) into the training process poses two challenges. Firstly, it is difficult for the model to capture the fine nuances of human perception. Secondly, it may significantly reduce the performance on other metrics, such as semantic alignment between motion and text. Therefore, optimizing for motion quality, text adherence, and human preferences jointly has been quite difficult in prior approaches.

In this work, we propose MotionRL, a reinforcement learning (RL)-based framework for multi-objective optimization in human motion generation. Unlike previous methods that often ignore human perception, our approach uses an RL framework (Schulman et al., 2017) to capture subtle and complex human perceptual priors from perception models. To avoid degradation of other metrics, we also incorporate motion quality and text adherence as part of the reward. However, as demonstrated in other fields (Lee et al., 2024), simple weighted combinations of rewards often lead to unstable training, especially when optimizing for complex human perception. To this end, we introduce a multi-reward optimization strategy to approximate Pareto optimality. Intuitively, each generated sample in a batch embodies a distinct trade-off among the three rewards, with some samples exhibiting better trade-offs than others. Instead of updating gradients using all batch samples, our approach selects non-dominated points that have better trade-offs. This allows the model to automatically learn the optimal balance among different rewards. Furthermore, our method learns reward-specific preference prompts, which can be used individually or in combination to control the trade-offs between rewards during inference, solving the problem of manually adjusting reward weights and ensuring more stable training.

In summary, our contributions are three-fold as follows:

- We propose MotionRL, a novel reinforcement learning-based multi-reward optimization algorithm for fine-tuning human motion generation. It effectively optimizes multiple rewards, including text adherence, motion quality, and human preferences, and allows the use of reward-specific tokens during inference to control the output.

- We make the first attempt to introduce reinforcement learning for incorporating and improving human perception in the text-to-motion domain. Unlike previous methods that either ignore human perception or directly incorporate perceptual priors into training, our RL-based approach is more effective at capturing complex human perceptions and produces results that are more convincing than those evaluated by error-based metrics.

- We evaluate our approach with experimental results, which demonstrate the superiority of our approach across traditional metrics like FID and R-Precision, as well as human perceptual model scores and user studies.

## 2 RELATED WORK

### 2.1 HUMAN MOTION GENERATION

Many works have attempted to generate motion from text. Recent mainstream research has employed stochastic models for motion generation. For instance, the diffusion method (Zhang et al., 2024; Chen et al., 2023; Tevet et al., 2022) describes motion generation as a denoising process conditioned on text. Some BERT-type masked generative models (Guo et al., 2023) have also successfully tackled this task. However, these approaches face a significant limitation: they require the motion length as input to guide generation, which can lead to significant motion quality degradation (Pinyoanuntapong et al., 2024). In contrast, GPT-type generative models (Zhang et al., 2023a; Guo et al., 2022c) solve this issue by first learning a VQVAE (Van Den Oord et al., 2017) to map continuous motion into discrete motion tokens, and models like Zhang et al. (2023a) and Mao et al. (2024) employ a transformer decoder to perform next token prediction.

### 2.2 HUMAN FEEDBACKS OF GENERATED MOTION

Human evaluation of generated motion should be highly correlated with motion quality and is arguably more important than traditional metrics such as FID or R-Precision(Heusel et al., 2017). However, in the text-to-motion domain, few studies incorporate human perceptual priors. Recent works, such as Voas et al. (2023), have established datasets of human motion ratings, and others, like Wang et al. (2024), use contrastive methods to improve the robustness of human evaluations. However, these approaches directly integrate ratings into the training process, which can negatively impact other metrics and fail to capture subtle human perceptual differences.

### 2.3 REINFORCEMENT LEARNING WITH HUMAN FEEDBACKS

Reinforcement Learning from Human Feedback (RLHF) (Nguyen et al., 2017) has achieved tremendous success in fine-tuning generative models, whether for text generation(OpenAI, 2024) or image generation(Fan et al., 2024). RL effectively aligns model outputs with human evaluations and offers significant advantages over explicit alignment methods for complex optimization objectives. OpenAI's technical report (OpenAI, 2024) noted that fine-tuning pretrained GPT models with RL helps models better understand human perceptions. However, in the text-to-motion domain, few studies use RL to align with human perceptions. For example, Mao et al. (2024) uses RL to guide generation, but without incorporating any human perceptual priors, focusing only on enhancing text-to-motion alignment.

## 3 PRELIMINARY

### 3.1 AUTOREGRESSIVE MODEL FOR MOTION GENERATION

A common method (Zhang et al., 2023a) to design a motion generator is using vector quantized variational autoencoder (VQ-VAE) (Van Den Oord et al., 2017) and generative pre-trained transformer (GPT). In T2M-GPT, the trained VQ-VAE maps motion sequences $m = [x_1, x_2, \ldots, x_T]$ to

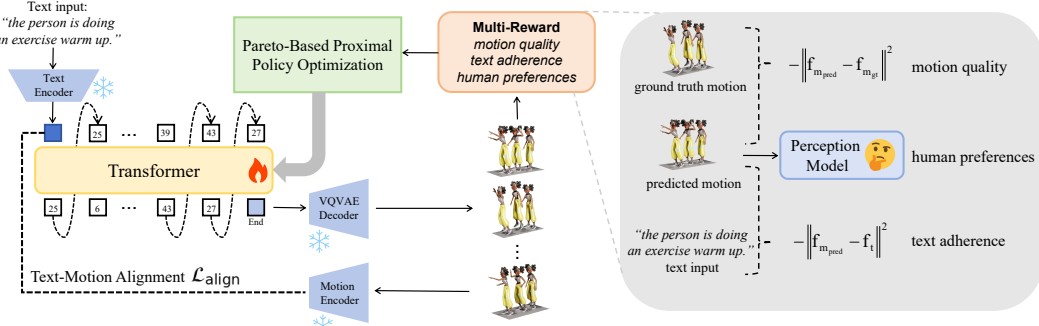

Figure 2: **The overall pipeline of MotionRL.** Given a text input, the Transformer serves as a motion generator, first producing multiple motions as a batch. Various rewards are then computed for these motions. Within this batch of motions, the Pareto set is identified. Finally, using the rewards from the Pareto set, along with the outputs of the critic model and the prediction logits, the motion generator is optimized using the PPO algorithm (note that the critic model is omitted in the diagram).

a sequence of indices $S = [s_1, s_2, \ldots, s_{T/l}, End]$ which are indices from the learned codebook $C$. Note that a special $End$ token is added to indicate the end of a motion. The VQ-VAE uses a standard CNN-based architecture and is trained following the details in T2M-GPT (Zhang et al., 2023a).

With a trained VQ-VAE, the motion generation task can be described as a next-token prediction task. In this stage a transformer is designed to autoregressively predict the next index $s_i$ based on previous indices set $S_{<i}$ and text condition $c$. The optimization goal in this stage is to maximize the distribution $p(S|c) = \prod_{i=1}^{|S|} p(s_i|c, S_{<i})$. We train the GPT with cross-entropy loss $\mathcal{L}_{CE}$.

Some relevant works introduce a sequence-level semantic supervision to help motion-text alignment. Specificlly, pre-trained motion encoders and text encoders like TMR encoders (Lu et al., 2023) can measure the similarity between the input text embedding and output motion embedding. The text embedding from the encoder aligns better with human motions than that from CLIP (Radford et al., 2021) or other LLMs. It also provides a way to directly supervise the text-motion alignment. So we calculate an additional loss $\mathcal{L}_{align}$ for supervising the alignment.

### 3.2 REWARD-SPECIFIC TOKEN DESIGN AND SAMPLING

Lin et al. (2022) introduced the use of preference information in multi-objective optimization. In MotionRL, reward-specific identifiers are employed, which allow the model to differentiate between various types of rewards during multi-objective optimization. The details about sampling and token design and are in the Appendix B.

## 4 METHODOLOGY

Most existing methods for text-to-motion generation are based on supervised training on pre-existing datasets, followed by performance evaluation on test sets. However, human perception of the quality of motion generation is often more meaningful than the performance metrics obtained from test sets. Human feedback signals, however, are more complex compared to standard evaluation metrics like FID (Heusel et al., 2017), and incorporating them directly into training poses challenges such as overfitting and the inability to dynamically optimize model outputs (Fan et al., 2024). To address these issues, we propose the MotionRL, which utilizes RL to fine-tune the model using human perception data. Furthermore, since incorporating perception introduces multiple optimization objectives, rather than using traditional methods of reward averaging, we adopt a Pareto-based multi-reward optimization approach as shown in Figure 2.

In Section 3.1, we first explain how MotionRL reformulates the text-to-motion task as an autoregressive process, which serves as the foundation for applying RL fine-tuning. We then proceed to the design of multi-objective rewards for RL, detailed in Section 4.1. For a given text input, multiple

samples can be generated, each associated with several rewards. To address this multi-objective optimization challenge, MotionRL introduces a batch-wise Pareto-optimal selection strategy, discussed in Section 4.2. Finally, Section 4.3 provides a detailed explanation of our Pareto-based policy gradient optimization method.

## 4.1 MULTI-REWARD DESIGN

Regarding the design of the reward models, we define three specific objectives to fine-tune the model's output: text adherence, motion quality, and human preferences.

For text adherence and motion quality, we use the paired text encoder and motion encoder from Guo et al. (2022a) and Lu et al. (2023). The training objective of the encoder in Guo et al. (2022a) is:

$$\mathcal{L}_{CL} = y\left(\|\mathbf{f}_t - \mathbf{f}_m\|^2\right) + (1-y)\left(\max(0, m - \|\mathbf{f}_t - \mathbf{f}_m\|)\right)^2, \tag{1}$$

where $\mathbf{f}_t$ and $\mathbf{f}_m$ denotes the embeddings extracted by text and motion encoders, respectively. $y$ is a binary label indicating the matched text-motion pairs and $m$ is the manually set margin.

The training objective of the encoder in Lu et al. (2023) is:

$$\mathcal{L}_{InfoNCE} = -\frac{1}{B}\sum_{i=1}^{B}\left[\log\frac{\exp\left(\mathbf{f}_t^i \cdot \mathbf{f}_m^i/\tau\right)}{\sum_{j=1}^{B}\exp\left(\mathbf{f}_t^i \cdot \mathbf{f}_m^j/\tau\right)} + \log\frac{\exp\left(\mathbf{f}_m^i \cdot \mathbf{f}_t^i/\tau\right)}{\sum_{j=1}^{B}\exp\left(\mathbf{f}_m^i \cdot \mathbf{f}_t^j/\tau\right)}\right], \tag{2}$$

where $\tau$ is learnable temperature parameter and $B$ is the batch size.

Now we have pretrained motion encoders and text encoders, we can define the text adherence reward as:

$$r_t = -\sum^{i}\lambda_i\|\mathbf{f}_{t,i} - \mathbf{f}_{m_{\text{pred}},i}\|^2, \tag{3}$$

where $\mathbf{f}_{m_{\text{pred}}}$ is the generated motion contioned on $t$ , $i$ represents different encoders and $\lambda$ represents their respective weights, used to constrain the values to a similar magnitude. Higher rewards indicating a better match between the generated motion and the input text.

Similarly, the motion quality reward is defined as:

$$r_m = -\sum^{i}\lambda_i\|\mathbf{f}_{m_{\text{gt}},i} - \mathbf{f}_{m_{\text{pred}},i}\|^2, \tag{4}$$

where $m_{\text{gt}}$ is the ground truth motion sequence. Higher rewards signify that the generated motion is closer to the ground truth.

We also need a model to align the human preferences. Giving an input motion sequence $m$, an implicit perception model $\mathcal{P}$ is assumed, where a higher rate indicates that the motion has better quality. To align the generated motions with human perception, we need a computational perception model $\mathcal{C}$ that best aligns $\mathcal{P}$. We use the model from Wang et al. (2024) as our human perception model. The pairwise comparison annotations from the collected dataset $\mathcal{D}$ can be used to calculate the training loss:

$$\mathcal{L}_{perception} = -\mathbb{E}_{(m^{(h)},m^{(l)})\sim\mathcal{D}}[\log\sigma(\mathcal{C}(m^{(h)}), \mathcal{C}(m^{(l)}))], \tag{5}$$

where $m^{(h)}$ is the better motion and $m^{(l)}$ is the worse one. After that, the model $\mathcal{C}$ is able to map the high-dimension motion to a reward $r_p$ as the motion rating. Therefore the human preference reward is defined as:

$$r_p = \mathcal{C}(g(m_{\text{pred}})), \tag{6}$$

where $g$ is a function that converts human motion from a 3D coordinate parameterized form to an SMPL (Loper et al., 2015) parameterized form. Here we train a simple neural network to achieve this goal rather than traditional methods. Please refer the Appendix A.

We normalized three different types of rewards to constrain them within the same order of magnitude. For specific normalization methods, please refer to Appendix C.

## 4.2 BATCH-WISE PARETO-OPTIMAL SELECTION

Lin et al. (2022) pointed out that using batchwise Pareto-set learning and selecting good samples in a batch can approximate Pareto-optimality across multiple objectives. In the preliminary stage, for $k$ types of rewards, we generated $k$ texts with special tokens, and each text sampled $N$ motions respectively. Therefore, MotionRL will find the corresponding Pareto-optimal set in this $N$ batch denoted as $\mathcal{M}$. For the different sampled motions $m_i$, the dominance relationship is defined as follows:

Let $m_a, m_b \in \mathcal{M}$, $m_a$ is considered to dominate $m_b$, denoted as $m_a \succ m_b$, if and only if $r_k(m_a) \geq r_k(m_b), \forall k \in \{1, \ldots, K\}$ and $\exists j \in \{1, \ldots, m\}$ such that $r_j(m_a) > r_j(m_b)$.

In order to enable the model to approximate Pareto optimality during the training process, we designed the algorithm as Algorithm 1.

---

**Algorithm 1:** MotionRL: Pareto-optimal Multi-Reward RL for Motion Generation

---

**Input:** Text $t$, Batch size $N$, Total iteration $E$, the number of rewards: $K$, Motion generation
   model: $\pi_\theta$, reference model: $\pi_{\text{ref}}$, Total sampling steps $T$
**for** $e = 1$ *to* $E$ **do**
   Sample text prompt $t \sim \pi(t)$;
   **for** $k = 1$ *to* $K$ **do**
      Prepend reward-specific tokens to $t$, obtain $t_k$;
      Sample a set of motions $\{m_1, \ldots, m_N\} \sim \pi_\theta(m|t_k)$;
      A reward vector $\mathbf{r}_k = \{r_{k1}, \ldots, r_{kN}\}$;
   Initialize empty non-dominated set $\mathcal{P}$;
   **for** $i = 1$ *to* $N$ **do**
      Initialize flag $dominated \leftarrow$ False;
      **for** $j = 1$ *to* $N$ and $i \neq j$ **do**
         **if** $\forall k, r_{ki} \leq r_{kj}$ *and* $\exists k, r_{ki} < r_{kj}$ **then**
            $dominated \leftarrow$ True;
            break;
      **if** $dominated =$ *False* **then**
         Add motion $m_i$ to non-dominated set $\mathcal{P}$;
   $\mathcal{J}_r(\pi_\theta) = \mathbb{E}_{t \sim p_{data}, m \sim \pi_\theta} \left[ \sum_{k=1}^{K} \frac{1}{n(\mathcal{P})} \sum_{i=1, m_i \in \mathcal{P}}^{N} \left[ r(t_k, m_k) - \beta \log \frac{\pi_\theta(m_k|t_k)}{\pi_{\text{ref}}(m_k|t_k)} \right] \right]$.
   Update $\pi_\theta$ using Proximal Policy Optimization (PPO);
**Output:** Fine-tuned motion generation model $\pi_\theta$.

---

## 4.3 PARETO-BASED POLICY GRADIENT OPTIMIZATION

We employ reinforcement learning to optimize the alignment between motion sequences and textual descriptions, human perception preferences, and motion quality. A Pareto-based multi-reward objective guides this process, balancing the different rewards.

The actor model $\pi_\theta$ generates motion sequences by selecting motion tokens at each time step based on the input text, while the critic model $V_\phi(s_t)$ estimates the value of the current state. The objective function is:

$$\mathcal{J}_r(\pi_\theta) = \mathbb{E}_{t \sim p_{data}, m \sim \pi_\theta} \left[ \sum_{k=1}^{K} \frac{1}{n(\mathcal{P})} \sum_{i=1, m_i \in \mathcal{P}}^{N} \left[ r(t_k, m_k) - \beta \log \frac{\pi_\theta(m_k \mid t_k)}{\pi_{\text{ref}}(m_k \mid t_k)} \right] \right], \quad (7)$$

where $r(t_k, m_k)$ represents the multi-objective reward function evaluating the alignment between the textual description $t_k$ and the generated motion sequence $m_k$, taking into account factors like semantic consistency, human preference, and motion quality. $\pi_\theta(m_k \mid t_k)$ is the actor model that predicts the motion $m_k$ based on the input text $t_k$, and $\pi_{\text{ref}}(m_k \mid t_k)$ is the reference model serving as a constraint to regulate how much the updated policy can deviate from the pre-trained model. The term $\beta$ controls the strength of this regularization.

We employ Proximal Policy Optimization (PPO) (Schulman et al., 2017) for training, where the advantage function is computed as:

$$A_t = G_t - V_\phi(s_t). \tag{8}$$

Here, $A_t$ measures the relative improvement of the selected action at time $t$ compared to the expected value. $G_t$ denotes the return, or the total reward accumulated from time step $t$, and $V_\phi(s_t)$ is the value function estimated by the critic model, representing the expected return at state $s_t$.

The actor's loss function is given by:

$$\mathcal{L}_{\text{actor}}(\theta) = \mathbb{E}_t \left[ \min \left( r_t(\theta) A_t, \text{clip}(r_t(\theta), 1 - \epsilon, 1 + \epsilon) A_t \right) \right], \tag{9}$$

where the clipping function ensures that the policy update is constrained to prevent large deviations from the previous policy. The critic model is updated by minimizing the squared error between the predicted value and the actual return:

$$\mathcal{L}_{\text{critic}}(\phi) = \mathbb{E}_t \left[ (V_\phi(s_t) - G_t)^2 \right]. \tag{10}$$

This framework allows both models to effectively learn from multiple reward signals, balancing semantic alignment, human preferences, and motion quality in the generated sequences.

## 5 EXPERIMENTS

### 5.1 EXPERIMENT SETTING

We select InstructMotion as our baseline model. The text-to-motion transformer is composed of 18 layers, each with a hidden size of 1,024 and 16 attention heads. For the PPO algorithm, we use a mini-batch size of 32 during training. We run the PPO algorithm for 2 epochs, using the AdamW (Loshchilov & Hutter, 2017) optimizer with $\beta_1 = 0.9$ and $\beta_2 = 0.99$. The pre-trained text-to-motion generator is fine-tuned over 40k iterations, with a learning rate of 5e-6. All experiments are performed on 4 NVIDIA RTX 3090 GPUs. The specific method for reward normalization is detailed in Appendix C, while the process and results of the user study are elaborated in Appendix D.

### 5.2 QUANTITATIVE EVALUATION

Tables 1 provides an evaluation of our MotionRL framework on the widely used dataset: HumanML3D. The results indicate that MotionRL outperforms the baseline models, T2M-GPT and MoMask, by a significant margin in key quantitative metrics such as R-Precision and FID. This demonstrates that the motion sequences generated by our model exhibit stronger alignment with the corresponding textual descriptions and better quality of motion. Diversity evaluates the diversity of generated motions across the entire test set, while modality assesses the diversity of motions generated from the same text. Compared to others, these are not core metrics.

It is important to note that $\S$ indicates reliance on the ground-truth sequence length for generation. All methods that depend on the ground-truth motion length tend to perform better on the FID metric because they ensure that the motion lengths are consistent with the test set prior to inference. However, Pinyoanuntapong et al. (2024) points out that superior FID scores for these methods do not necessarily imply higher motion quality. To demonstrate the superiority of MotionRL, we utilize the scores from the motion perception model provided in Wang et al. (2024) alongside real human user study ratings.

Table 1: **Quantitative comparison on HumanML3D test set.** The evaluation metrics are computed following Guo et al. (2022b). § indicates reliance on ground-truth sequence length for generation. Underline indicates the second best. The closer Diversity is to the ground truth, the better.

| Methods | R-Precision↑ | | | FID↓ | MM-Dist↓ | Diversity | MModality↑ |
|---|---|---|---|---|---|---|---|
| | Top-1 | Top-2 | Top-3 | | | | |
| **Ground truth motion** | 0.511 | 0.703 | 0.797 | 0.002 | 2.974 | 9.503 | - |
| TEMOS§ (Petrovich et al., 2022) | 0.424 | 0.612 | 0.722 | 3.734 | 3.703 | 8.973 | 0.532 |
| MLD§ (Chen et al., 2023) | 0.481 | 0.673 | 0.772 | 0.473 | 3.196 | 9.724 | 2.192 |
| MDM§ (Tevet et al., 2022) | - | - | 0.611 | 0.544 | 5.566 | 9.559 | 1.907 |
| MotionDiffuse§ (Zhang et al., 2024) | 0.491 | 0.681 | 0.782 | 0.630 | 3.113 | 9.410 | 0.730 |
| GraphMotion§ (Jin et al., 2024) | 0.504 | 0.699 | 0.785 | 0.116 | 3.070 | 9.692 | **2.766** |
| ReMoDiffuse§ (Zhang et al., 2023b) | 0.510 | 0.698 | 0.795 | 0.103 | 2.974 | 9.018 | 1.239 |
| MoMask§ (Guo et al., 2023) | 0.521 | 0.713 | 0.807 | **0.045** | 2.958 | - | 1.131 |
| (Ahuja & Morency, 2019) | 0.246 | 0.387 | 0.486 | 11.02 | 5.296 | - | - |
| Ghosh et al. (2021) | 0.301 | 0.425 | 0.552 | 6.532 | 5.012 | - | - |
| TM2T (Guo et al., 2022c) | 0.424 | 0.618 | 0.729 | 1.501 | 3.467 | 8.589 | 2.424 |
| Guo et al. (2022b) | 0.455 | 0.636 | 0.736 | 1.087 | 3.347 | 9.175 | 2.219 |
| T2M-GPT (Zhang et al., 2023a) | 0.491 | 0.680 | 0.775 | 0.116 | 3.118 | 9.761 | 1.856 |
| Fg-T2M (Wang et al., 2023) | 0.492 | 0.683 | 0.783 | 0.243 | 3.109 | 9.278 | 1.614 |
| MotionGPT (Jiang et al., 2023) | 0.492 | 0.681 | 0.778 | 0.232 | 3.096 | **9.528** | 2.008 |
| InstructMotion (Mao et al., 2024) | 0.505 | 0.694 | 0.790 | 0.099 | 3.028 | 9.741 | - |
| **Ours** | **0.531** | **0.721** | **0.811** | 0.066 | **2.898** | 9.653 | 1.385 |

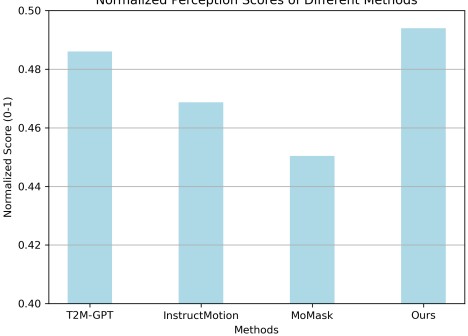

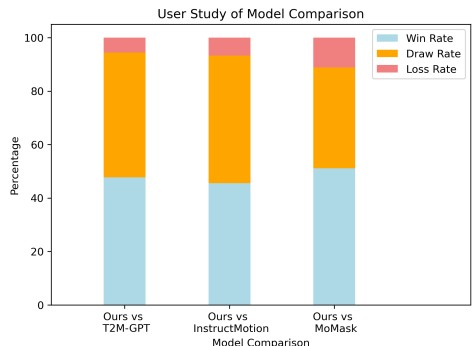

(a) Perception model scores of different methods      (b) User Study of Model Comparison

Figure 3: **Human Preferences Evaluation.** (a) Perceptual scores on the test set using the pretrained perception model from Wang et al. (2024). The results show that our method aligns more closely with human perception compared to other approaches. (b) Comparison of human evaluations between our method and others. The results demonstrate that our method generates motions that are more consistent with human preferences.

Figure 3(a) illustrates the output scores from the motion perception model in Wang et al. (2024). It is evident that our model achieves higher perceptual scores compared to other models, indicating that it effectively captures human preference information embedded in the perceptual model, leading to motions that align more closely with human perception. Figure 3(b) presents the scores given by real human evaluators for the model-generated motions. We compared the success rates of MotionRL against other models, further demonstrating the overall superiority of our model in terms of motion quality.

## 5.3 QUALITATIVE EVALUATION

To validate the superiority of MotionRL in generating high-quality actions, we compared MotionRL with other well-performing models, including T2M-GPT, InstructMotion, and Momask, based on

| T2M-GPT | InstructMotion | MoMask | **Ours** |
|---|---|---|---|

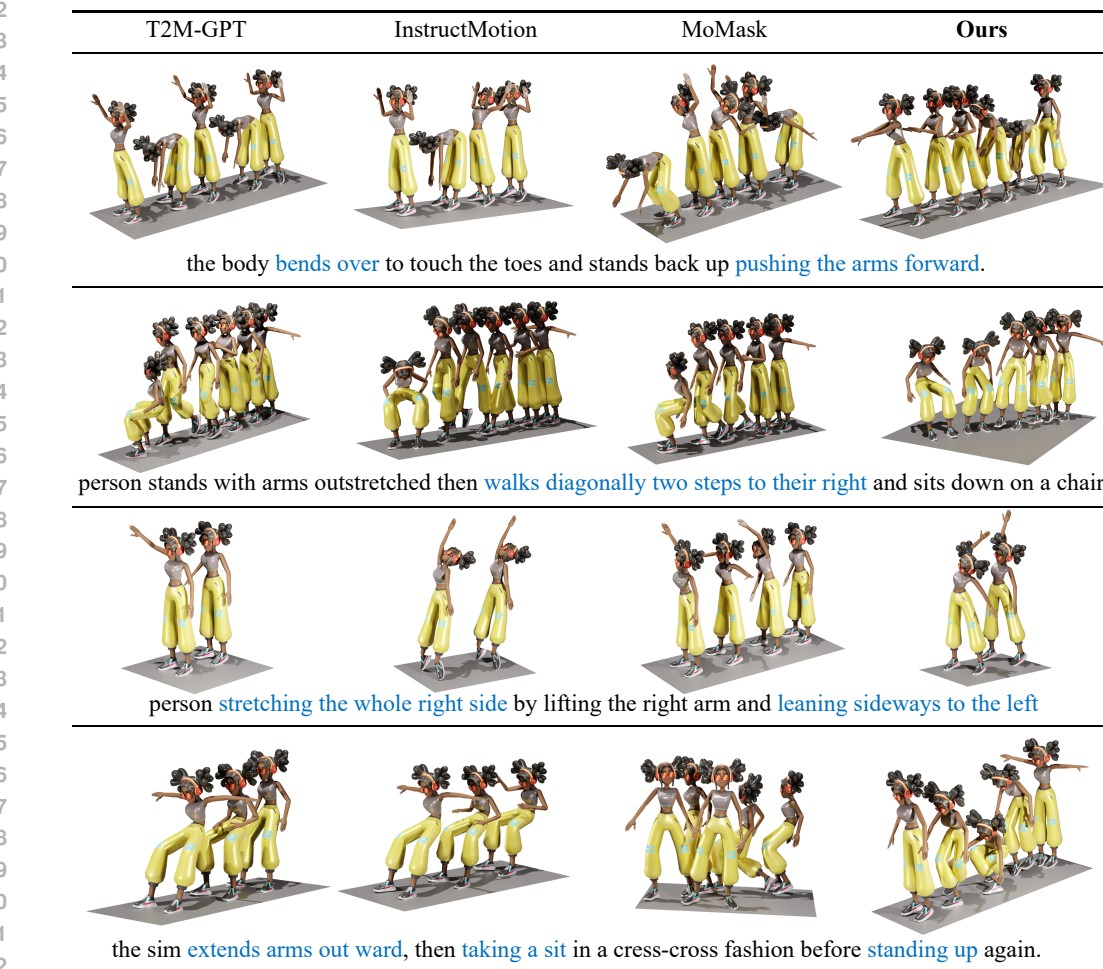

the body bends over to touch the toes and stands back up pushing the arms forward.

person stands with arms outstretched then walks diagonally two steps to their right and sits down on a chair.

person stretching the whole right side by lifting the right arm and leaning sideways to the left

the sim extends arms out ward, then taking a sit in a cress-cross fashion before standing up again.

Figure 4: **Qualitative comparisons with top-performing methods.** Our MotionRL exhibits better motion generation quality.

several prompts from the test set. As shown in Figure 4, our method generates actions that align with the text, whereas the other methods fail to produce accurate actions. Our method produces high-quality motions that correspond to the text. More details can be found in the supplementary materials.

## 5.4 ABLATION

**Reward Design**: Our framework employs a multi-objective optimization approach with three distinct rewards. To verify that our method effectively captures human perceptual preferences, we conducted numerical experiments on the HumanML3D test set. By using different reward combinations, we assessed our method's performance in terms of both traditional metrics and human perceptual model scores. We observed that the perception reward we designed effectively enhanced the output of the human perception model. Additionally, the rewards we developed for motion quality and text adherence significantly improved performance on both FID and Top-1 Precision metrics.

**Pareto-based Optimization**: Instead of transforming the multi-objective optimization into a single reward through weighted summation, MotionRL approximates Pareto optimality within each batch of samples. To verify the superiority of our method, we compared the effects of using batch-wise Pareto selection and different reward-specific tokens on model rewards. As shown in Figure 3, the use of Pareto optimization effectively improved the overall reward values of the model. Additionally,

Table 2: **Ablation of Reward Design** $R_p$ is the human preferences reward, $R_m$ is the motion quality reward, and $R_t$ is the text adherence reward. The results indicate that each of the designed rewards, along with our multi-objective optimization approach, successfully improves the generated motions in different aspects. Notably, when all three rewards are combined, the model achieves the best overall performance.

| $R_p$ | $R_m$ | $R_t$ | Top-1 $\uparrow$ | FID $\downarrow$ | Perception $\uparrow$ |
|---|---|---|---|---|---|
| $\checkmark$ | | | 0.519 | 0.090 | **0.495** |
| | $\checkmark$ | $\checkmark$ | 0.528 | **0.064** | 0.465 |
| $\checkmark$ | $\checkmark$ | $\checkmark$ | **0.531** | **0.064** | 0.494 |

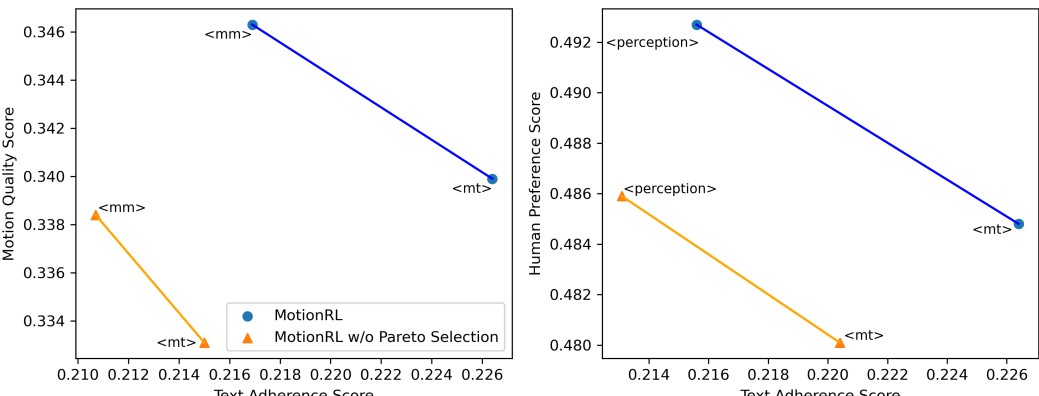

Figure 5: **Impact of Pareto Selection and Reward-Specific Tokens.** $<$mt$>$, $<$mm$>$, and $<$perception $>$represent reward-specific tokens corresponding to text adherence, motion quality, and human preferences, respectively. It illustrates the effectiveness of our proposed Pareto selection in enhancing the model's overall reward value. It also demonstrates how using different reward-specific tokens allows for trade-offs between various optimization goals.

employing different reward-specific tokens allowed effective control over the model's output. It is important to note that comparisons of reward values across different optimization objectives are meaningless because our normalization process during training constrains the rewards to the same scale. However, the physical meaning of the values for different objectives varies, and only the relative magnitudes of rewards within the same optimization objective are meaningful.

## 6 CONCLUSION AND FUTURE WORK

We propose MotionRL, an algorithmic framework for generating human motions based on GPT and reinforcement learning. Addressing the complexities and challenges of capturing human perception, we draw inspiration from other fields by integrating existing human motion perception models with reinforcement learning. This innovative approach uniquely tackles the alignment of human perception with generated motions, an area where no other methods currently exist. To enhance motion generation quality and textual alignment, we introduce two additional rewards in our reinforcement learning framework. Rather than employing traditional reward-weighted averaging, we propose a batch-wise Pareto sample selection optimization method. Evaluations of both quantity and quality demonstrate significant success in human perception, motion quality, and text alignment. The discussion about limitation and future work is shown in Appendix F.

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

## A    CONVERT JOINTS DATA TO SMPL

To efficiently convert joint-based motions to SMPL format without the computational overhead of iterative methods, we developed a lightweight neural network. The traditional approach required over 20 iterations per sequence, making it impractical for real-time training in our framework. Our goal was to significantly reduce conversion time while maintaining the quality of the generated SMPL motions.

The network consists of a combination of 1D convolutional layers (Conv1D) and Long Short-Term Memory (LSTM) units:

**Conv1D Layers**: These layers capture spatial dependencies between joints within each frame.

**LSTM Layers**: LSTMs are used to model the temporal dynamics of the motion sequences, allowing the network to understand how motions evolve over time.

**Fully Connected Output Layer**: Finally, a fully connected layer converts the processed features into SMPL format, where each frame consists of 25 joints with 6 parameters (rotation data). This architecture allows the network to efficiently handle the transformation from joint-based data to SMPL format, leveraging the strong temporal and spatial relationships in the data.

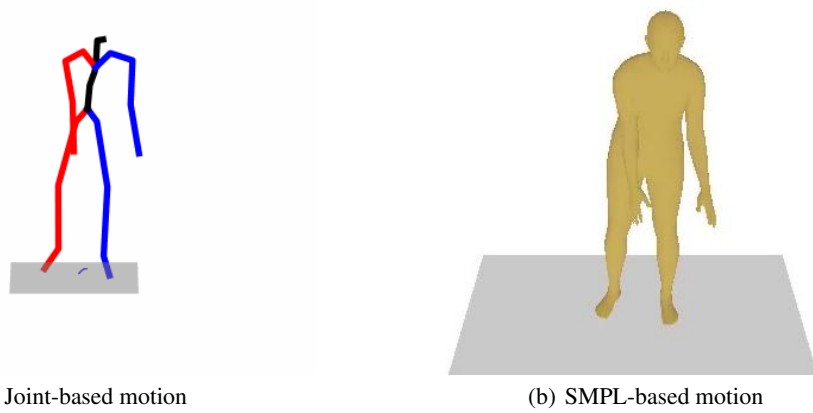

(a) Joint-based motion                     (b) SMPL-based motion

Figure 6: **Visualization of motions in different formats** (a) Original joint-based motion (b) SMPL-based motion after conversion using our trained model

After training on the HumanML3D dataset, using SMPL motions generated through iterative methods as ground truth, the model demonstrated both speed and accuracy in converting joint-based motions into SMPL format. The neural network performed the conversion much faster than traditional methods, making it feasible for real-time use during model training. This enabled us to integrate SMPL format conversion directly into the training loop, optimizing both the efficiency and the overall quality of the generated motions.

## B  DETAILS OF SAMPLING AND TOKEN DESIGN

This section provides a detailed explanation of our sampling strategy and the design of reward-specific tokens in the training process.

In our implementation of the Proximal Policy Optimization (PPO) algorithm, the actor model is a fine-tuned Transformer that generates the probability distribution for predicting the next motion index. The selection of the next index from this probability distribution follows a sampling strategy.

**Sampling Strategy** During training, for a given text prompt $t$, multiple samples $n$ are generated. Common approaches include probability-based sampling or beam search (greedy sampling). However, in practice, we found that greedy sampling tends to cause the critic model to overfit, meaning the actor model lacks sufficient exploration. To address this, we adopted probability-based sampling with a temperature coefficient set to 1.5. This encourages broader exploration by the actor model, leading to more diverse and varied motion outputs during training.

**Reward-Specific Token Design** The text inputs to the model are not simply raw text but include a special token at the end of each sentence. For example, the prompt *"a person is running forward"* is modified to *"a person is running forward <mm>"*, where *<mm>* signifies that the motion quality reward is being calculated for this sample. Similarly, the token *<mt>* indicates a text adherence reward, and *<perception>* represents the human preference reward.

This token-based approach allows the model to differentiate between different reward types, guiding the model towards optimizing multiple objectives. However a key challenge in the text-to-motion domain is that the dataset size is significantly smaller compared to fields like image or text generation. This makes the model more sensitive to small changes in input text. Directly introducing these reward-specific tokens into the input text led to a noticeable drop in performance during initial training, as the pre-trained model had no prior knowledge of these tokens and thus struggled to interpret them.

To address this issue, we employed weighted guidance by combining both the original text and the modified reward-specific text during training.

$$\hat{\mathbf{f}}_{t_k} = (1 - \alpha)\mathbf{f}_t + \alpha\mathbf{f}_{t_k}, \tag{11}$$

where $\alpha$ is a weight parameter, $t$ is original texts , $t_k$ is the texts with reward-specific tokens and $\mathbf{f}$ is the text encoder. By adjusting the weight of the features corresponding to these special tokens, we ensure that the model's output is not overly influenced by the tokens themselves. At the same time, the model retains the ability to distinguish between different reward types, something that traditional single-objective optimization approaches are unable to achieve.

This approach effectively balances exploration and reward differentiation, allowing the model to generate high-quality motions while accounting for multiple optimization goals.

## C  REWARD NORMALIZATION

In this section, we explain how we normalize different reward values for stable training.

Since we already employ reward-specific tokens, different rewards produce slightly different text features even when the input text remains the same. This prevents the model from confusing different reward inputs, even without directly weighting or summing the rewards. However, the token's ability to control the output might be limited. To ensure more stable training, we normalize all rewards to the same scale.

We use an extended min-max normalization method:

$$r_{k,\text{normalized}} = \begin{cases} \frac{r_k - \text{min\_val}_k}{\text{max\_val}_k - \text{min\_val}_k}, & \text{if min\_val}_k \leq r_k \leq \text{max\_val}_k \\ \frac{r_k - \text{min\_val}_k}{\text{max\_val}_k - \text{min\_val}_k}, & \text{if } r_k < \text{min\_val}_k \\ \frac{r_k - \text{max\_val}_k}{\text{max\_val}_k - \text{min\_val}_k} + 1, & \text{if } r_k > \text{max\_val}_k \end{cases} \tag{12}$$

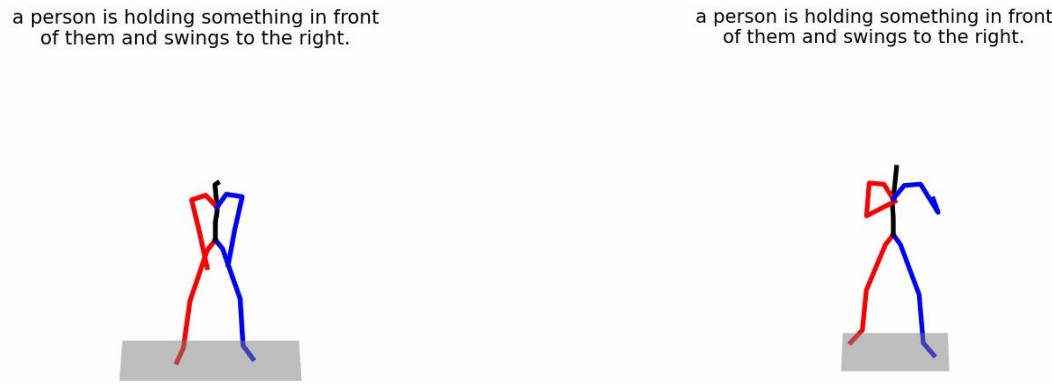

Figure 7: **An example page displayed to volunteers.** These GIFs are randomly shuffled.

In this equation, $\text{min\_val}_k$ and $\text{max\_val}_k$ represent the estimated minimum and maximum values for each reward type $k$. These estimates do not need to be highly precise because, even if the reward values exceed the expected range slightly—whether a bit over 1 or below 0—it does not affect the stability of training.

It is important to note that the normalized rewards across different reward types are not directly comparable. Our goal is simply to bring all rewards to the same scale, without needing to precisely control the normalization range or fine-tune weight parameters as required by traditional weighted-sum methods. This flexibility is enabled by our use of reward-specific tokens and the Pareto-based policy gradient.

## D  DETAILS OF USER STUDY

**Data Preparation:** We randomly selected 30 prompts from the HumanML3D test set. Each prompt describes a specific human motion.

**Model Inference:** Using various models, we generated corresponding human motions based on the same prompts, and the generated motions were rendered as GIF images.

**Evaluation Method:** To compare the performance of our model with other models, we presented the GIFs created by different models to volunteers for evaluation. For each set of GIFs, we recruited 4-6 volunteers to assess the overall quality of the motions. The volunteers were asked to choose which of the two motions better matched the text description, exhibited higher quality, and appeared smoother and more natural. Figure 7 shows an example of the interface we provided to the volunteers, where the order of the GIFs was randomized.

Volunteers could select one of three options for each comparison: which motion was better (win or loss), or indicate if it was too difficult to decide (draw). We then analyzed these responses to determine the performance of each model.

## E  MORE QUALITATIVE VISUALIZATIONS

In our supplementary materials, we provide additional motion examples that showcase the superiority of our method in terms of text adherence, motion quality, and human preference. These examples demonstrate how our approach outperforms others, not only based on quantitative metrics but also in real human perception.

## F  LIMITATION AND FUTURE WORK

While our model effectively captures complex human perceptual information using existing perception model output scores and reinforcement learning, we have not introduced additional human

annotation costs prior to training, relying solely on pre-trained motion perception models. This fine-tuning approach is heavily dependent on the quality of the perception models themselves.

To further enhance the quality of human perception, we have the goal of developing a user-friendly interface in the future. This will allow for the real-time collection of real human feedback and further model adjustments. We also referenced OpenAI's well-known work, InstructGPT(Ouyang et al., 2022), in our paper. InstructGPT applies the idea of RLHF (Reinforcement Learning with Human Feedback) to align large language models with human preferences based on real-time feedback, which also inspired MotionRL and our ongoing research. We believe that with real-time human feedback data, we will significantly address the current limitations of perception model quality and improve the quality of generated motions.

