# OpenReview forum: "MotionRL: Align Text-to-Motion Generation to Human Preferences with Multi-Reward Reinforcement Learning"
_ICLR.cc/2025/Conference — ICLR 2025 Conference Withdrawn Submission_

### Official Review · Reviewer_hqhg · 2024-11-01

**Soundness:** 2
**Presentation:** 1
**Contribution:** 2
**Rating:** 5
**Confidence:** 4

**Summary:**

This paper introduces MotionRL, a reinforcement learning (RL)-based framework for multi-objective optimization in human motion generation. Recognizing that prior methods often overlook human perceptual nuances, MotionRL integrates human perceptual priors into the generation process through a reward structure. The framework balances three objectives: human perception, motion quality, and text adherence. To address the challenges of optimizing multiple objectives, MotionRL incorporates a multi-reward optimization strategy that approximates Pareto optimality by selecting non-dominated points in each batch, ensuring better trade-offs across objectives.

**Strengths:**

The experimental metrics show good results, and a motion generation framework based on RL has been designed.

**Weaknesses:**

The reason for introducing RL is not well explained.

**Questions:**

1. In line 95, it seems that the purpose of introducing RL is merely to supervise human preference, and the rationale for using RL isn’t well explained.

2. The concept of Pareto optimality lacks a citation—what specific issue does this?

3. There are factual inaccuracies in the stated contributions, as ReinDiffuse appears to have introduced RL into motion before this paper.

4. The supplementary materials don’t clearly explain their purpose, which only provides some generated samples.

5. The paper contains several grammatical errors. The title should change "Align" to "Aligning". A layout issue in line 292.

---

> ### Author Response · Authors · 2024-11-19
> **Rebuttal by Authors**
>
> We are grateful for the insightful comments and clarify the concerns as follows.
>
> **Q1**: The reason for introducing RL is not well explained. In line 95, it seems that the purpose of introducing RL is merely to supervise human preference, and the rationale for using RL isn’t well explained.
>
> **A1**:
> We clarify the necessity and advantages of RL based on the experimental conclusions from this paper and other works:
> 1. According to OpenAI's famous work InstructGPT, using RL for aligning large language models with human preferences makes it **easier to capture complex preference information**. This inspired our idea of using RL in motion generation to align human preferences.
> 2. In the text-to-image field, many works have successfully and stably fine-tuned multiple rewards for generated images using RL, such as aesthetic scores and human preferences (e.g., DPOK). These works demonstrate that RL, compared to direct explicit supervision, is **more advantageous in handling complex reward structures and multi-objective optimization**. However in the field of motion generation, no previous work has used RL to align human preference priors.
> 3. Using RL supports unpaired labeled datasets as rewards. In motion generation tasks that align human preferences, other works have explicitly incorporated paired preference information into the training process. This brings significant data collection challenges, as it doesn't support real-time online data collection and fine-tuning. By using RL, we can **develop user-friendly motion generation models and collect real human feedback in real-time**, which will improve motion generation quality and reduce reliance on manually labeled data.
>
> **Q2**: The concept of Pareto optimality lacks a citation—what specific issue does this?
>
> **A2**: The concept of Pareto optimality is explained in the paper Pareto Set Learning that we referenced, but we did not include the citation at the point where the concept first appears. We have added it now.
> The complete definition of Pareto optimality is as follows:
>
> We consider the following continuous multi-objective optimization problem:
> $$
> \min_{m \in \mathcal{M}} r(m) = (r_1(m), r_2(m), \dots, r_K(m)),
> $$
> where $ m $ is a solution in the decision space $ \mathcal{M} \subset \mathbb{R}^n $, $ r : \mathcal{M} \rightarrow \mathbb{R}^K $ is a $ K $-dimensional vector-valued objective function.
>
> Let $ m_a, m_b \in \mathcal{M} $. $ m_a $ is said to dominate $ m_b $, denoted as $ m_a \succ m_b $, if and only if $ r_k(m_a) \geq r_k(m_b), \forall k \in \{1, \dots, K\} $ and $ \exists j \in \{1, \dots, K\} $ such that $ r_j(m_a) > r_j(m_b) $.
>
> **Definition (Pareto Optimality)** A solution $ m^* \in \mathcal{M} $ is *Pareto optimal* if there is no $ \hat{m} \in \mathcal{M} $ such that $ \hat{m} \succ m^* $.
>
> **Q3**: There are factual inaccuracies in the stated contributions, as ReinDiffuse appears to have introduced RL into motion before this paper.
>
> **A3**:
> We need to point out that ReinDiffuse was first publicly released on October 9, 2024, but the ICLR 2025 paper submission deadline was October 1, 2024. Therefore, the statement "ReinDiffuse appears to have introduced RL into motion before this paper" is inaccurate. We are still the first to introduce RL for aligning human perception in the motion generation field.
>
> Furthermore, ReinDiffuse does not use real human perception priors, but instead uses computed numerical values (e.g., foot contact) as reinforcement learning reward signals. These do not involve human feedback priors, so we can still claim that **we are the first to introduce RL to align human perception priors**. The significant contribution of our work lies in proposing a new motion generation framework that balances optimization across multiple objectives based on RL and pareto optimization.
>
> **Q4**: The supplementary materials don’t clearly explain their purpose, which only provides some generated samples.
>
> **A4**:
> To avoid violating the double-blind principle of the review process, we cannot provide the webpage used for our user study to display the results more intuitively. Therefore, we included some generated samples from our method in the supplementary materials to allow reviewers to visually inspect the motion generation results. Now we have added comparison results with the current state-of-the-art method, MoMask, and if you're interested, you can see the direct visual comparison.
>
> **Q5**: The paper contains several grammatical errors. The title should change "Align" to "Aligning". A layout issue in line 292.
>
> **A5**:
> Thank you for pointing this out. We have corrected the issue in the newly uploaded PDF.
>
> [1] Ouyang, Long, et al. "Training language models to follow instructions with human feedback." NeurIPS 2022
>
> [2] Fan, Ying, et al. "Reinforcement learning for fine-tuning text-to-image diffusion models." NeurIPS 2024
>
> [3] Lin, Xi, et al. "Pareto set learning for expensive multi-objective optimization." NeurIPS 2022

---

> > ### Comment · Reviewer_hqhg · 2024-11-25
> >
> > Thank you to the authors for their efforts in the rebuttal. I believe the rationale for introducing RL is not convincing enough. Many methods claim to 'capture complex preference information' as a similar advantage, but a deeper explanation of the underlying motivation for using this technique is needed. By the way, when Reindiffuse was first introduced, it was already stated that it had been accepted by the conference, so its actual release date is earlier than this paper.  I am more inclined to reject this paper, so I maintain my score.

---

### Official Review · Reviewer_aZMF · 2024-11-03

**Soundness:** 3
**Presentation:** 3
**Contribution:** 2
**Rating:** 5
**Confidence:** 4

**Summary:**

The paper offers MotionRL, an approach for optimizing text-to-motion generation tasks using Multi-Reward RL. This method aims to align the generated motions with human preferences more effectively than traditional models by incorporating human perceptual models into the RL training process. This is done by employing a multi-objective optimization strategy to balance text adherence, motion quality, and human preferences, approximating Pareto optimality. The paper provides experiments, demonstrating that the proposed model outperforms baselines on traditional metrics and in user studies assessing human perception.

**Strengths:**

* Introducing RL with the set of reward functions to optimize for multiple objectivesin text-to-motion generation seems to be a novel approach to that specific domain.

* The idea is simple and easy to understand which is also further improved by a good presentation and writing. The images helps me understand what is the problem you're trying to solve.

* I find the ablation in table 2 important as it shows that human perception does not always align with the metricses typically used.

**Weaknesses:**

* The overall novelty of the method is modest at best, but the domain application seem to be novel.

* There is a great lack of details about the user study and "volunteers for evaluation" could mean anyone. Given that the participants was called "volunteers" (and not paid?) it seems like it's not randomly recruited people which makes it hard to evaluate the soundness of the user study.

* The paper uses up all 10 pages but given that it's a simple idea with limited technical contribution I think it's excessive, especially since the limitation was pushed to the appendix.

**Questions:**

* How did you come up with the balance between the three different rewards?

* How is the pareto-based policy gradient optimization different from simple PPO?

* What is the demographic data for the user study, did you do any formal survey?

Minor things:
- You introduce the abbreviation "RL" multiple times in the text
- Line 200: the Appdix -> the Appendix

---

> ### Author Response · Authors · 2024-11-19
> **Rebuttal by Authors**
>
> We are grateful for the insightful comments and clarify the concerns as follows.
>
> **Q1**: What is the demographic data for the user study, did you do any formal survey?
>
> **A1**: We will provide more details about the user study.
>
> We randomly selected 30 prompts from the HumanML3D test set, performed inference using different models, and visualized the results as GIFs. Since we compared 3 groups, we ended up with a total of 90 pairs of GIFs. We then designed an anonymous survey webpage, where on each page, these paired GIFs (the order of the 90 pairs was completely shuffled, and the images within each pair were also randomized) were presented for users to choose the winner, draw, or loser for each pair. For the user study, we distributed the anonymous survey webpage on a volunteer platform at a university. Each user was required to select the preferred image for 30 pairs, so the 90 pairs were divided into 3 sets, and each set was evaluated by 4-6 anonymous volunteers. This means that **each prompt’s pairwise results were assessed by 4-6 volunteers, with a total of 16 different volunteers participating. Each volunteer evaluated 30 pairs of data**.
>
> During the evaluation process, we did not know the personal information of the volunteers, and the volunteers also did not know our personal information. The volunteers were not paid, as the anonymous webpage was distributed via the university's volunteer mutual support platform. However, **the use of a two-way anonymous recruitment process ensures the fairness of the user study**.
>
> **Q2**: How did you come up with the balance between the three different rewards?
>
> **A2**: We use **reward-specific tokens** to better handle the challenges of multi-objective optimization and trade-offs. During the inference phase, we append different special tokens to different text prompts. For example, **adding the <mm> token makes the model more inclined to output samples with a higher motion quality reward, while adding the <perception> token makes the model more inclined to output samples that align better with human perception**. This means that even if the input prompts are the same, users can choose which reward to emphasize, allowing the generated motions to balance the three rewards according to the user's preferences.
>
> **Q3**: How is the pareto-based policy gradient optimization different from simple PPO?
>
> **A3**: Compared to standard PPO, the Pareto-based policy gradient optimization is an improvement strategy we designed specifically for multi-objective optimization. The standard PPO algorithm updates the optimization by taking the weighted sum of multiple reward signals, whereas we use batch-wise Pareto selection. For each text input prompt, there are $n$ samples, and each sample has $k$ types of rewards. In the $n \times k$ matrix, we select the samples that lie on the Pareto frontier (the non-dominated set) to calculate the reinforcement learning optimization objective function. Additionally, for each different reward, we use reward-specific tokens to mark the text input.
>
> **Original PPO** (requires manual fine-tuning of reward weights, does not support trade-offs between rewards):
>
> $$
> \mathcal{J} _ r(\pi _ \theta) = \mathbb{E} _ {t \sim p _ {\text{data}}, m \sim \pi _ {\theta}} \left[ r(t, m) _ {\text{avg}} - \beta \log \frac{\pi _ \theta(m \mid t)}{\pi _ {\text{ref}}(m \mid t)} \right]
> $$
>
> Note that $r(t, m) _ {\text{avg}}$ requires fine-tuning the weights of multiple rewards and taking the average.
>
> **Pareto-based policy gradient optimization** (does not require manual fine-tuning of reward weights, supports trade-offs between different rewards during the inference phase, and performs better):
>
> $$
> \mathcal{J} _ r(\pi_\theta) = \mathbb{E} _ {t\thicksim p _ {data}, m\thicksim \pi _ {\theta}} \left[ \sum _ {k=1}^{K} \frac{1}{n(\mathcal{P})} \sum _ {i=1, m _ i \in \mathcal{P}}^{N} \left[ r(t _ k, m _ k) - \beta \log \frac{\pi _ \theta(m _ k \mid t _ k)}{\pi _ {\text{ref}}(m _ k \mid t _ k)} \right]\right]
> $$
>
> This strategy not only **improves the three rewards for generated motions overall**, but also **supports to make trade-offs during the inference phase based on user preferences**.
>
> **Q4**: The overall novelty of the method is modest at best, but the domain application seem to be novel.
>
> **A4**: From a motivational standpoint, we are the first to introduce RL for aligning human perception in the field of motion generation. Our reinforcement learning framework, PPO, is a classic method, but we have **designed three novel rewards specifically for the motion generation task**: text adherence, motion quality, and human preferences. This is one of the key innovations of our paper as **there is no previous works describe the motion generation task as a multi-objective optimization problem with RL**. Other works do not use RL, and they either focus solely on perception or on numerical performance, which prevents them from generating high-quality motions.

---

### Official Review · Reviewer_zcPF · 2024-11-04

**Soundness:** 2
**Presentation:** 2
**Contribution:** 2
**Rating:** 6
**Confidence:** 3

**Summary:**

The paper "MotionRL: Align Text-to-Motion Generation to Human Preferences with Multi-Reward Reinforcement Learning" introduces MotionRL, a framework that leverages Multi-Reward Reinforcement Learning to enhance text-to-motion generation by aligning outputs with human preferences. Unlike prior models, MotionRL incorporates human perceptual data in its reward structure to address limitations in previous approaches focused on numerical metrics alone. MotionRL employs a multi-objective optimization strategy to achieve a balance between text adherence, motion quality, and human preferences, approximating Pareto optimality for optimal trade-offs. The authors demonstrate the model's effectiveness through extensive experiments and user studies, showing MotionRL's superior performance in terms of perceptual quality and alignment with human expectations.

**Strengths:**

The paper introduces a novel approach by combining reinforcement learning with human preference alignment for text-to-motion tasks, which is an underexplored area in motion generation. The application of multi-reward RL to text-to-motion represents a good contribution, especially as it integrates human perceptual feedback.

**Weaknesses:**

The model’s reliance on pre-trained perception models, as mentioned in the limitations, could restrict generalizability. Fine-tuning without additional human annotations might limit the model’s adaptability to new or unique datasets, where the pre-trained perception model may not fully capture the nuances of human preferences.

While the multi-reward RL framework is effective, there is limited discussion on dynamically adjusting the weight of each reward in real-time based on specific user feedback. A more adaptive reward weighting mechanism could further enhance user-centered customization.

Although the human preference model provides valuable perceptual data, additional insights from continuous human interaction could help refine the model further. The paper would benefit from exploring how human feedback could be iteratively incorporated to improve long-term model performance, potentially through ongoing human-in-the-loop adjustments.

**Questions:**

Could the authors elaborate on the potential for dynamically adjusting the weighting of rewards based on real-time feedback?

How does the model handle scenarios where text descriptions are ambiguous or open to interpretation? Is there a mechanism to weigh certain rewards more heavily in such cases?

For practical applications, are there plans to develop more user-friendly interfaces for non-technical users to fine-tune MotionRL’s generated motions?

Please also provide some details about the dataset you created

---

> ### Author Response · Authors · 2024-11-19
> **Rebuttal by Authors**
>
> We are grateful for the insightful comments and clarify the concerns as follows.
>
> **Q1**: While the multi-reward RL framework is effective, there is limited discussion on dynamically adjusting the weight of each reward in real-time based on specific user feedback. A more adaptive reward weighting mechanism could further enhance user-centered customization.
>
> Could the authors elaborate on the potential for dynamically adjusting the weighting of rewards based on real-time feedback?
>
> How does the model handle scenarios where text descriptions are ambiguous or open to interpretation? Is there a mechanism to weigh certain rewards more heavily in such cases?
>
> **A1**: **The reward-specific tokens designed in our method allow for a trade-off between three different rewards, enabling user-centered customization during the inference phase**. This process is **real-time** and **training-free**, meaning that no fine-tuning of the model is required to adjust the generated results according to user preferences. In contrast, traditional reinforcement learning approaches to multi-objective optimization typically weight multiple reward signals into a single reward for optimization, which requires manual fine-tuning of reward weights and does not allow for choices to be made during inference.
>
> Our designed reward-specific token solves the trade-off between the three objectives. During inference, we append different special tokens to various text prompts. For example, **adding the <mt> token after the text makes the model more likely to output motions that adhere closely to the text description. This helps address the issue of motion generation not following the text when the description is vague.** Similarly, adding the <perception> token makes the model more likely to generate samples that align with human perception. This means that even with the same input prompt, users can choose which reward to emphasize.
>
> In our ablation study, we demonstrated the effects of using different tokens. It can be observed that using the corresponding token allows us to adjust the generated motion’s reward value, balancing the trade-off across the three aspects.
>
> **Our method also supports accepting user feedback and fine-tuning the model based on this feedback.** This is a significant advantage of reinforcement learning in aligning human preferences in generative tasks. The reinforcement learning objective function is as follows:
>
> $$
> \mathcal{J} _ r(\pi _ \theta) = \mathbb{E} _ {t\thicksim p _ {data}, m\thicksim \pi _ {\theta}} \left[ \sum _ {k=1}^{K} \frac{1}{n(\mathcal{P})} \sum _ {i=1, m _ i \in \mathcal{P}}^{N} \left[ r(t _ k, m _ k) - \beta \log \frac{\pi _ \theta(m _ k \mid t _ k)}{\pi_{\text{ref}}(m _ k \mid t _ k)} \right]\right]
> $$
>
> In the formula, we incorporate the real-time feedback weak-label data collected as part of the reward value $r(t _ k, m _ k)$ (which needs to be adjusted based on the user’s confidence level). This allows for real-time adjustment of the overall model performance. **Currently, our method can fine-tune the motion generators based on user feedback data without any modification**, and collecting and publicly releasing feedback data will be part of our future work.
>
> **Q2**: Although the human preference model provides valuable perceptual data, additional insights from continuous human interaction could help refine the model further. The paper would benefit from exploring how human feedback could be iteratively incorporated to improve long-term model performance, potentially through ongoing human-in-the-loop adjustments.
>
> For practical applications, are there plans to develop more user-friendly interfaces for non-technical users to fine-tune MotionRL’s generated motions?
>
> **A2**: Yes, we have adopted a reinforcement learning strategy to enable fine-tuning of the model, with **the goal of developing a user-friendly interface in the future**. This will **allow for the real-time collection of authentic human feedback and further model adjustments**. We also referenced OpenAI's well-known work, InstructGPT, in our paper. InstructGPT applies the idea of RLHF (Reinforcement Learning with Human Feedback) to align large language models with human preferences based on real-time feedback, which also inspired MotionRL and our ongoing research. Your suggestion is very insightful and aligns with our future plans. We believe that **with real-time human feedback data, we will significantly address the current limitations of perception model quality**.
>
> **Q3**: Please also provide some details about the dataset you created.
>
> **A3**: Currently, we have not created a new dataset. The dataset used for training and evaluation is HumanML3D, and the dataset used for the perception model is MotionCritic. However, in future work, we plan to collect real-time human feedback data to optimize the performance of both the perception model and the motion generator.

---

### Official Review · Reviewer_N7W1 · 2024-11-08

**Soundness:** 3
**Presentation:** 3
**Contribution:** 2
**Rating:** 6
**Confidence:** 3

**Summary:**

The paper introduces a novel approach that leverages Multi-Reward Reinforcement Learning to optimize text-to-motion generation tasks. Unlike previous methods that primarily focus on numerical performance metrics, MotionRL incorporates human preferences to enhance the alignment of generated motions with human perception. The approach uses a multi-objective optimization strategy to balance text adherence, motion quality, and human preferences, aiming for Pareto optimality. Extensive experiments and user studies demonstrate that MotionRL significantly outperforms existing methods in generating high-quality, realistic motions that align well with textual descriptions and human feedback. This work represents a significant advancement in the field of text-driven human motion generation, offering a more nuanced and human-centric approach to evaluating and improving motion quality.

**Strengths:**

Human-Centric Optimization: MotionRL uniquely incorporates human preferences into the optimization process, ensuring that the generated motions align better with human perception. This focus on human feedback addresses the limitations of traditional metrics that may not fully capture the nuances of human motion quality.
Multi-Objective Optimization: The use of a multi-reward reinforcement learning framework allows MotionRL to balance multiple objectives simultaneously, such as text adherence, motion quality, and human preferences. This approach ensures that the generated motions are not only accurate and high-quality but also meet the subjective preferences of users.
Pareto Optimality: MotionRL introduces a novel multi-objective optimization strategy to approximate Pareto optimality. By selecting non-dominated points within each batch, the model learns to balance different rewards effectively, leading to more stable and optimal training outcomes. This method enhances the overall performance across various metrics compared to other algorithms.

**Weaknesses:**

Dependence on Pre-trained Perception Models: MotionRL relies on pre-trained human perception models to capture complex human perceptual information. If these models are not of high quality or do not accurately reflect human preferences, the performance of MotionRL could be significantly impacted. This dependency limits the flexibility and robustness of the approach.

Limited Dataset Size: The text-to-motion domain typically has smaller datasets compared to other fields like image or text generation. This limitation makes the model more sensitive to small changes in input text and can lead to overfitting. The smaller dataset size also poses challenges in effectively training the model to generalize well across diverse motion scenarios.

Complexity of Multi-Objective Optimization: While the multi-reward optimization strategy aims to balance text adherence, motion quality, and human preferences, it introduces significant complexity into the training process. Managing and fine-tuning multiple rewards can lead to unstable training and requires careful calibration to ensure that the model does not prioritize one objective at the expense of others. This complexity can make the approach less accessible and harder to implement effectively.

**Questions:**

How does MotionRL handle the trade-offs between text adherence, motion quality, and human preferences during the training process?

What are the limitations of relying on pre-trained human perception models for aligning generated motions with human preferences, and how can these limitations be addressed in future work?

---

> ### Author Response · Authors · 2024-11-19
> **Rebuttal by Authors**
>
> We are grateful for the insightful comments and clarify the concerns as follows.
>
> **Q1**: Dependence on Pre-trained Perception Models: MotionRL relies on pre-trained human perception models to capture complex human perceptual information. If these models are not of high quality or do not accurately reflect human preferences, the performance of MotionRL could be significantly impacted. This dependency limits the flexibility and robustness of the approach.
>
> What are the limitations of relying on pre-trained human perception models for aligning generated motions with human preferences, and how can these limitations be addressed in future work?
>
> **A1**: Currently, MotionRL does indeed rely heavily on the quality of the perception model. However, we have adopted a reinforcement learning strategy to enable fine-tuning of the model, with **the goal of developing a user-friendly interface in the future**. This will **allow for the real-time collection of authentic human feedback and further model fine-tuning**.
>
> Supporting online real-time fine-tuning of the model is one of the key advantages of reinforcement learning. Previously, there has been no work in the field of motion generation that uses reinforcement learning to align with human perception. This is why our approach has great potential to significantly improve the human perception of generated motions. We also referenced OpenAI's well-known work, InstructGPT, in our paper. InstructGPT applies the idea of RLHF (Reinforcement Learning with Human Feedback) to align large language models with human preferences based on real-time feedback, which also inspired MotionRL and our ongoing research. We believe that **with real-time human feedback data, we will significantly address the current limitations of perception model quality**.
>
> **Q2**: Complexity of Multi-Objective Optimization: While the multi-reward optimization strategy aims to balance text adherence, motion quality, and human preferences, it introduces significant complexity into the training process. Managing and fine-tuning multiple rewards can lead to unstable training and requires careful calibration to ensure that the model does not prioritize one objective at the expense of others. This complexity can make the approach less accessible and harder to implement effectively.
>
> How does MotionRL handle the trade-offs between text adherence, motion quality, and human preferences during the training process?
>
> **A2** : MotionRL uses **reward-specific tokens** to better handle multi-objective optimization and trade-off issues. During the inference stage, we append different special tokens to the input prompts, such as **adding the <mt> token after the text to make the model tend to generate motions that better match the user’s text description**, or adding the <perception> token to make the model tend to generate motions that align more closely with human perception. This means that even with the same input prompt, the user can choose which reward they want to prioritize, allowing the generated motion to balance the three rewards according to the user's preference. This design enables **effective trade-offs between multiple objectives without the need for fine-tuning the reward weights**. The trade-off is both **user-centric** and **real-time**.
>
> Besides we employ batch-wise Pareto selection, which selects samples on the Pareto frontier across the three reward dimensions within each batch. Our ablation experiments have demonstrated that **this strategy improves the three rewards for generated motions**.
>
> **Q3**: Limited Dataset Size: The text-to-motion domain typically has smaller datasets compared to other fields like image or text generation. This limitation makes the model more sensitive to small changes in input text and can lead to overfitting. The smaller dataset size also poses challenges in effectively training the model to generalize well across diverse motion scenarios.
>
> **A3**: The size of the dataset is indeed closely related to the quality of motion generation. In our paper, we used the most common dataset in this field, HumanML3D, which contains 14,616 motion segments totaling 28.6 hours. Based on experimental results and experience, for generating motions under 10 seconds or motions from shorter text inputs, our method is not highly sensitive to small changes in the input text, unless the input text is extremely long. The main contribution of our method is proposing an effective framework to optimize the human perception preferences, text adherence, and motion quality of generated motions. If larger datasets become available in the future, we will expand our method to work with those larger datasets.

---

### Official Review · Reviewer_qQod · 2024-11-08

**Soundness:** 2
**Presentation:** 3
**Contribution:** 2
**Rating:** 3
**Confidence:** 4

**Summary:**

The paper proposes incorporating human preference priors into the text-to-motion task to enhance the quality of generated motions. Through the use of reinforcement learning and a motion perception model, the paper construct a human preference reward, enabling the generation model to learn human perception. To prevent degradation in other performance metrics resulted by human preference reward, the paper introduces a motion quality reward and a text adherence reward, forming a proposed multi-reward system. To mitigate potential training instability caused by multiple rewards, Pareto optimality is employed to balance the different rewards. The experimental results demonstrates superior performance in FID, R-Precision, human perceptual model scores, and user studies.

**Strengths:**

This paper introduces human preference into the text-to-motion generation task by applying a motion perception model from the motion generation field. Additionally, two other rewards (text adherence and motion quality) are used to prevent potential degradation caused by the human preference reward. Experimental results demonstrate the superiority of the proposed approach.

**Weaknesses:**

The most concerning limitation of this paper lies in its novelty. Introducing human perception into the field of motion generation is not novel, as also mentioned by this paper in line#86-87 (Voas et al., 2023; Wang et al., 2024). Wang et al. utilizes their proposed motion perception model as an effective supervision signal to finetune the motion generator. This paper similarly uses the motion perception model from Wang et al. as a supervision signal to train model for motion generation. Given these, the technical novelty is limited.

There are also insufficiencies in the experimental section: **1)** The main insufficiency is that the ablation study does not adequately demonstrate the effectiveness of the proposed multiple rewards. Table 2 lacks a baseline ablation result with no rewards applied, making it difficult to confirm the method’s effectiveness. **2)** In text-to-motion generation tasks, methods are typically validated on both the HumanML3D and KIT-ML datasets, but this paper only provides results for HumanML3D. **3)** Additionally, on the HumanML3D dataset, the MModality metric is usually reported; however, this paper neither provides MModality results nor explains why it was omitted. **4)** For the Diversity metric, Diversity closer to real test data is preferable, yet this paper labels higher values as better.

Minor comments:
1. The paper does not explain the metrics reported in the tables, such as Diversity.
2. The “Conclusion and Future Work” section does not include future work but instead links to the appendix.
3. Descriptions for figures should not be placed in the appendix; for example, <mm> and <mt> in Fig. 5 lack explanation in the main text.

**Questions:**

1. “Aligning Motion Generation with Human Perceptions” by Wang et al., 2024, utilizes their proposed motion perception model as an effective supervision signal for training a motion generator. This paper similarly uses the motion perception model from Wang et al.  as a supervision signal to aid motion generation. What are the distinction and novelty of this paper's method compared to the approach of Wang et al.?
2. The use of **skeleton** rendering in the user study section for evaluation could be more reasonable, as there is a gap between **skeleton** and **SMPL** rendering. Some unnatural details are more noticeable in SMPL, using SMPL rendering for the user study might be more convincible.

---

> ### Author Response · Authors · 2024-11-21
> **Rebuttal by Authors (1/2)**
>
> We are grateful for the insightful comments and clarify the concerns as follows.
>
> **Q1**: “Aligning Motion Generation with Human Perceptions” by Wang et al., 2024, utilizes their proposed motion perception model as an effective supervision signal for training a motion generator. This paper similarly uses the motion perception model from Wang et al. as a supervision signal to aid motion generation. What are the distinction and novelty of this paper's method compared to the approach of Wang et al.?
>
> **A1**:
>
> 1. The methods proposed by Voas et al., 2023, and Wang et al., 2024 use human perceptual signals as a single, direct guide, which leads to a significant drop in both motion quality and text alignment. After fine-tuning with a single perceptual signal, the generated motions are smoother but no longer match the text. In contrast, our method is **the first to treat motion generation as a multi-objective optimization, considering multiple aspects of the motion**, which makes it more balanced than previous approaches.
>
> 2. In tasks like text-to-image generation, similar work (e.g., InstructGPT, NeurIPS 2022) has shown that **reinforcement learning performs better than direct supervision in capturing complex perceptual signals**, especially when aligning perceptual signals with other inputs. This advantage of reinforcement learning has been proven in other areas as well, such as DPOK (NeurIPS 2024).
>
> 3. Voas et al., 2023, and Wang et al., 2024 treat perceptual signals as explicit supervision and train a fixed perceptual model, requiring retraining whenever new data is collected. In contrast, we use the perceptual model output as a reinforcement learning reward, allowing for online fine-tuning. This means that after deployment, the model can **adjust in real-time using user perceptual data, improving both the perceptual model and motion generation**, similar to RLHF in large language models. We plan to focus on this approach in future work, collecting perceptual data to further refine the model.
>
> **Q2**: The use of skeleton rendering in the user study section for evaluation could be more reasonable, as there is a gap between skeleton and SMPL rendering. Some unnatural details are more noticeable in SMPL, using SMPL rendering for the user study might be more convincible.
>
> **A2**: Your suggestion is very reasonable. Indeed, using skeleton-based animation loses some of the finer details in the motion, which could explain the higher draw rate we observed in our user study. If we use SMPL-based animation, users would likely be able to better distinguish the quality of the motion, so we expect the draw rate to decrease, while the win and loss rates should increase. Considering the clear distinction in win and loss rates from our user study, where MotionRL outperforms other methods, we believe that using skeleton animation for the user study does not affect the conclusion of our paper. We will switch to using SMPL animation in future work.

---

> ### Author Response · Authors · 2024-11-21
> **Rebuttal by Authors (2/2)**
>
> **Q3**: There are also insufficiencies in the experimental section: 1) The main insufficiency is that the ablation study does not adequately demonstrate the effectiveness of the proposed multiple rewards. Table 2 lacks a baseline ablation result with no rewards applied, making it difficult to confirm the method’s effectiveness. 2) In text-to-motion generation tasks, methods are typically validated on both the HumanML3D and KIT-ML datasets, but this paper only provides results for HumanML3D. 3) Additionally, on the HumanML3D dataset, the MModality metric is usually reported; however, this paper neither provides MModality results nor explains why it was omitted. 4) For the Diversity metric, Diversity closer to real test data is preferable, yet this paper labels higher values as better.
>
> **A3**: Thank you for pointing out the shortcomings in our experiment.
>
> 1. We indeed overlooked including a baseline ablation result with no rewards applied. This is easy to obtain, as we can skip the reinforcement learning phase and directly evaluate the model. The complete results are as follows:
>
> | $R_p$ | $R_m$ | $R_t$ | Top-1 $\uparrow$ | FID $\downarrow$ | Perception $\uparrow$ |
> |-------|-------|-------|------------------|------------------|-----------------------|
> |       |       |       | 0.520            | 0.097            |                0.468                 |
> | ✔     |       |       | 0.519            | 0.090            |                **0.495**             |
> |       | ✔     | ✔     | 0.528            | **0.064**        |              0.465                 |
> | ✔     | ✔     | ✔     | **0.531**        | **0.064**        |             0.494                 |
>
> The results show that the three types of rewards we designed effectively **improve the model's text adherence, motion quality, and human preferences**. This also supports our conclusion: **better FID does not necessarily indicate better human perception**.
>
> 2. We did not evaluate on the KIT-ML dataset because HumanML3D has more than three times the number of motion segments, making it more challenging to train on. Additionally, previous work, including the HumanML3D paper, shows that using the KIT-ML dataset for text-to-motion leads to poorer generalization and higher risk of overfitting. Other methods, like HumanTOMATO (ICML 2024), also did not use KIT-ML. Therefore, we believe evaluations on HumanML3D are more convincing than on KIT-ML.
>
> 3. The value of MModality is not the core focus or advantage of our paper. Other papers, like Motion-X (NeurIPS 2023), also do not consider MModality as a key factor, viewing it only as a supplementary reference metric. Taking your feedback into account, we have now included the results as follows:
>
> | Methods                          | Top-1  $\uparrow$   | Top-2  $\uparrow$   | Top-3  $\uparrow$   | FID$\downarrow$ | MM-Dist$\downarrow$ | Diversity | MModality$\uparrow$ |
> |---|---|--|--|---|--|---|---|
> | **Ground truth motion**  | 0.511| 0.703| 0.797| 0.002 | 2.974 | 9.503| - |
> | **TEMOS$^\S$** | 0.424  | 0.612| 0.722| 3.734 | 3.703 | 8.973| 0.532 |
> | **MLD$^\S$**       | 0.481  | 0.673| 0.772| 0.473| 3.196| 9.724  | 2.192  |
> | **MDM$^\S$**         | - | - | 0.611| 0.544 | 5.566 | *9.559* | 1.907|
> | **MotionDiffuse$^\S$**| 0.491| 0.681 | 0.782| 0.630| 3.113| 9.410| 0.730|
> | **GraphMotion$^\S$**  | 0.504| 0.699| 0.785| 0.116| 3.070| 9.692| **2.766**|
> | **ReMoDiffuse$^\S$**  | 0.510| 0.698| 0.795| 0.103| 2.974| 9.018| 1.239 |
> | **MoMask$^\S$**  | *0.521* | *0.713* | *0.807* | **0.045**       | *2.958*| - | 1.131 |
> | **Language2Pose**  | 0.246 | 0.387     | 0.486     | 11.02| 5.296| -  | - |
> | **Ghosh et al.**   | 0.301| 0.425 | 0.552     | 6.532| 5.012| -| - |
> | **TM2T**          | 0.424     | 0.618     | 0.729     | 1.501| 3.467  | 8.589     | *2.424*|
> | **Guo et al.**            | 0.455     | 0.636     | 0.736     | 1.087| 3.347| 9.175     | 2.219|
> | **T2M-GPT**   | 0.491     | 0.680     | 0.775     | 0.116 | 3.118| 9.761     | 1.856 |
> | **Fg-T2M**   | 0.492     | 0.683     | 0.783     | 0.243 | 3.109| 9.278     | 1.614               |
> | **MotionGPT** | 0.492     | 0.681     | 0.778     | 0.232 | 3.096| **9.528** | 2.008|
> | **InstructMotion**  | 0.505     | 0.694     | 0.790     | 0.099 | 3.028| 9.741     | -  |
> | **Ours**                | **0.531** | **0.721** | **0.811** | *0.066* | **2.898**| 9.653     | 1.385|
>
>
> 4. The way Diversity is measured varies across different papers. For example, T2M-GPT (CVPR 2023) and HumanTOMATO (ICML 2024) consider higher Diversity as better, while Motion-X (NeurIPS 2023) suggests that Diversity should be closer to real test data. Some papers, like MoMask (CVPR 2024), do not consider Diversity important and do not report it. Diversity is not the primary focus of our method and does not affect the main conclusions of our paper. Based on your feedback, we have updated the measurement to indicate that higher Diversity is better when it is closer to the real test data as above.

---

> ### Comment · Reviewer_qQod · 2024-11-25
>
> Thanks for the authors' response. Considering the following issues, the rating is adjusted to 3:
> - The two key technical aspects of this paper—human perception and reinforcement learning—lack sufficient innovation.
> - The experiments are inadequate and fail to effectively support the claims of innovation.
>
> Specifically,
>
> **Issues with insufficient innovation** in the human perception and RL components:
> 1. **Human Perception**:
>     1. **Misleading writing**: In the abstract, lines #15-17, the statement “Previous works … human feedback” misleads the reader into thinking that this paper’s contribution lies in introducing human feedback. However, applying human feedback to human motion was already proposed by Wang et al. In fact, this paper proposes to replace the previous direct using of human perception model with RL, rather than introduce human feedback into human motion area.
>     2. **Improvement in the Human Perception Score comes from previous work, not this paper’s innovations**:
>         1. While the authors emphasize the Perception Score and dedicate significant portions of the text and experiments to proving the rationale of human preference, the improvement in the Perception Score originates from the human perception model itself, as shown in the ablation table.
>          2. Furthermore, the human perception model used in this paper and the importance of human perception/preference were already proposed in Wang et al.’s work. Considering this, the contribution of this paper in the area of human perception is more about further validating Wang et al.'s work rather than demonstrating novelty.
>
> 2. **Reinforcement Learning**:
>     - **Lack of experimental support for the necessity of RL**:
>     Regarding the reasons and innovations behind using RL, although the authors respond that “reinforcement learning performs better than direct supervision in capturing complex perceptual signals,” further experimental results to substantiate this claim should be provided. For example, a comparison between "the current RL-based approach with 3 rewards" v.s. "using the 3 rewards directly as a loss function without RL" could clarify this point. However, the experimental section of this paper lacks evidence supporting the rationality and necessity of replacing the direct use of the human perception model with RL, which reduces the paper’s claims' overall persuasiveness.
>
> **Issues with insufficient experiments**:
>
> 1. This paper **only conducts experiments on the HumanML3D dataset, whereas other methods are typically validated on at least two datasets**. As a result, the findings of this paper are insufficient to fully support its conclusions and the effectiveness of its proposed method. Although the authors mentioned in their response that KIT-ML is less convincing and that HumanTOMATO also did not experiment on KIT-ML, it is worth noting that HumanTOMATO conducted experiments not only on the HumanML3D dataset but also on the larger-scale Motion-X dataset. In contrast, this paper validates its method solely on the HumanML3D dataset without providing results on other datasets, which undermines the credibility of the work.
> 2. Furthermore, while the authors claim that KIT-ML is not reliable, many works have demonstrated consistent performance across the HumanML3D and KIT-ML datasets, such as BAD, BAMM, and MoMask. It indicates that KIT-ML is a valuable dataset for evaluating models.
>     1. BAD: Bidirectional Auto-regressive Diffusion for Text-to-Motion Generation
>     2. BAMM: Bidirectional Autoregressive Motion Model
>     3. MoMask: Generative Masked Modeling of 3D Human Motions
> 3. The issue of insufficient experimental data was also highlighted by Reviewer N7W1.

---

### Author Response · Authors · 2024-11-26
**Withdrawal of Our Submission**

We would like to express our gratitude to all the reviewers for the time and effort they have invested in reviewing our work. Overall, we are pleased that Reviewers aZMF, N7W1, and zcPF recognize the contribution of our research.

We are confident in the value of our work, and we have diligently addressed the reviewers' concerns to the best of our ability. However, due to some negative feedback from certain reviewers, we have made the decision to withdraw our submission.

Specifically, we must express our concern regarding Reviewer qQod's unfair criticism. We have repeatedly emphasized in our paper that ours is **the first work to use reinforcement learning (RL)** to align generated motions with human perception. We have also highlighted the differences and advantages of our approach compared to Wang et al.'s method, and we included ablation experiments that demonstrate the effectiveness of RL. Despite being the first work in the field to consider optimization across multiple objectives, and achieving significant improvements over previous work in both numerical metrics and human perception, our paper was still questioned on the grounds of "lack of innovation" for using RL and relatively insignificant experimental details.

We also express our frustration with Reviewer hqhg, whose criticism included factual errors regarding the publication dates of other papers.

In conclusion, we once again thank all the reviewers for their efforts, and we will take their feedback into consideration in our future work.

---

### Note · Authors · 2024-11-26

I have read and agree with the venue's withdrawal policy on behalf of myself and my co-authors.